

# Litter inputs and standing stocks in riparian zones and streams under secondary forest and managed and abandoned cocoa agroforestry systems

Haialla Carolina Rialli Santos Brandão[1], Camila Andrade Coqueiro Moraes[2], Ana Paula Silva[2], José Francisco Gonçalves Júnior[3], Renan de Souza Rezende[4] and Daniela Mariano Lopes da Silva[5]

[1] Programa de Pós Graduação em Desenvolvimento e Meio Ambiente, Universidade Estadual de Santa Cruz, Ilhéus, Bahia, Brazil
[2] Universidade Estadual de Santa Cruz, Ilhéus, Bahia, Brazil
[3] AquaRiparia/Lab. de Limnologia, Departamento de Ecologia, Universidade de Brasília, Brasilia, Brazil
[4] Universidade Comunitária da Região do Chapecó, Chapecó, Santa Catarina, Brazil
[5] Departamento de Ciências Biológicas, Universidade Estadual de Santa Cruz, Ilhéus, Bahia, Brazil

Corresponding author
Daniela Mariano Lopes da Silva, dmlsilva@uesc.br

## ABSTRACT

**Background:** Cocoa is an important tropical tree crop that is mainly cultivated in agroforestry systems (AFS). This system, known as cabruca in northeastern Brazil, holds promise to reconcile biodiversity conservation and economic development. However, since cocoa AFS alters forest structure composition, it can affect litter dynamics in riparian zones and streams. Thus, our objective was to determine litter inputs and standing stocks in riparian zones and streams under three types of forest: managed cocoa AFS, abandoned cocoa AFS, and secondary forest.

**Methods:** We determined terrestrial litter fall (TI), vertical (VI) and lateral (LI) litter inputs to streams, and litter standing stocks on streambeds (BS) in the Atlantic Forest of northeastern Brazil. Litter was collected every 30 days from August 2018 to July 2019 using custom-made traps. The litter was dried, separated into four fractions (leaves, branches, reproductive organs, and miscellaneous material) and weighed.

**Results:** Terrestrial litter fall was similar in all forests, ranging from 89 g m$^{-2}$ month$^{-1}$ in secondary forest (SF) to 96 g m$^{-2}$ month$^{-1}$ in abandoned cocoa AFS (AC). Vertical input were higher in AC (82 g m$^{-2}$ month$^{-1}$) and MC (69 g m$^{-2}$ month$^{-1}$) than in SF (40 g m$^{-2}$ month$^{-1}$), whereas lateral input were higher in MC (43 g m$^{-2}$ month$^{-1}$) than in AC (15 g m$^{-2}$ month$^{-1}$) and SF (24 g m$^{-2}$ month$^{-1}$). Standing stocks followed the order SF > AC > MC, corresponding to 425, 299 and 152 g m$^{-2}$. Leaves contributed most to all litter fractions in all forests. Reproductive plant parts accounted for a larger proportion in managed AFS. Branches and miscellaneous litter were also similar in all forests, except for higher benthic standing stocks of miscellaneous litter in the SF. Despite differences in the amounts of litter inputs and standing stocks among the forests, seasonal patterns in the abandoned AFS (AC) were more similar to those of the secondary forest (SF) than the managed AFS, suggesting potential of abandoned AFS to restore litter dynamics resembling those of secondary forests.

# INTRODUCTION

Riparian zones are important for the functioning of headwater streams (*Vannote et al., 1980*; *Naiman, Décamps & Mcclain, 2005*), including in tropical zones (*Gonçalves Júnior et al., 2014*; *Bambi et al., 2016*; *Rezende et al., 2017a*; *Rezende et al., 2019*; *Calderón et al., 2019*). The riparian canopy limits instream primary production and provides allochthonous organic matter to stream and riparian food webs in the form of litter, which increases heterotrophic metabolism (*Gonçalves Júnior et al., 2014*; *Rezende et al., 2019*). Therefore, litter dynamics are a fundamental characteristic of headwater streams (*Abelho & Graça, 1996*; *Neres-Lima et al., 2017*). In the tropics, litter is typically supplied throughout the year (*Tonin et al., 2017*), although this pattern varies among forest types (*Lindman et al., 2017*; *Seena et al., 2017*), largely driven by precipitation and temperature regimes (*Bambi et al., 2016*; *Tonin et al., 2017*). Changes in the structure and composition of riparian forests can affect the supply of litter to streams and their riparian zones (*Delong & Brusven, 1994*; *Ferreira et al., 2019*; *Wild, Gücker & Brauns, 2019*), as well as in-stream litter dynamics (*Sutfin, Wohl & Dwire, 2016*; *Tiegs et al., 2019*).

Many tropical forests are jeopardized by rapid deforestation and expansion of agriculture (*Bawa et al., 2004*). This includes the Atlantic Forest of Brazil as one of the most threatened tropical forests worldwide (*Winbourne et al., 2018*; *Taubert et al., 2018*). Agroforestry systems (AFSs), however, have potential to partly reconcile the conservation of tropical forest patches with economic development (*Cassano et al., 2009*; *Schroth et al., 2011*). One example is the cultivation of cocoa in the Atlantic Forest of northeast Brazil where cocoa trees (*Theobroma cacao* L.) are grown in AFS that cover a large portion of the remnant Atlantic Forest (*Piasentin, Saito & Sambuichi, 2014*). The cocoa trees are planted in the shade of native forest trees (dominant and codominant strata) and are surrounded by natural vegetation. Therefore, cocoa AFS are thought to cause less environmental impact than other crop systems (*Johns, 1999*; *Sambuichi, 2002*), with benefits for local biodiversity (*Faria et al., 2007*; *Cassano et al., 2009*; *Schroth et al., 2011*). Important processes such as the incorporation of large amounts of organic matter into the forest soil are indeed maintained in cocoa AFS (*Beer et al., 1998*; *Gama-Rodrigues et al., 2010*; *Barreto et al., 2011*; *Fontes et al., 2014*; *Costa et al., 2018*). Nevertheless, changes in the vegetation structure of AFS compared to unmanaged forest may affect the amount of litter deposited in riparian zones (*Delong & Brusven, 1994*; *Wild, Gücker & Brauns, 2019*) and supplied to streams (*Gonçalves Júnior et al., 2014*).

Studies on litter dynamics in tropical streams and their riparian zones are scarce, especially under cocoa AFS, although some evidence suggests that replacing cocoa AFS changes the cycling of carbon and nitrogen in streams (*Costa et al., 2017*; *Souza et al., 2017*; *Costa et al., 2018*), possibly as a result of altered litter supply by riparian vegetation. Thus, the current study aimed to assess the influence of cocoa AFS on litter dynamics by determining differences in secondary forest and managed and abandoned AFS on litter inputs and benthic standing stocks in streams and riparian zones in these forests.
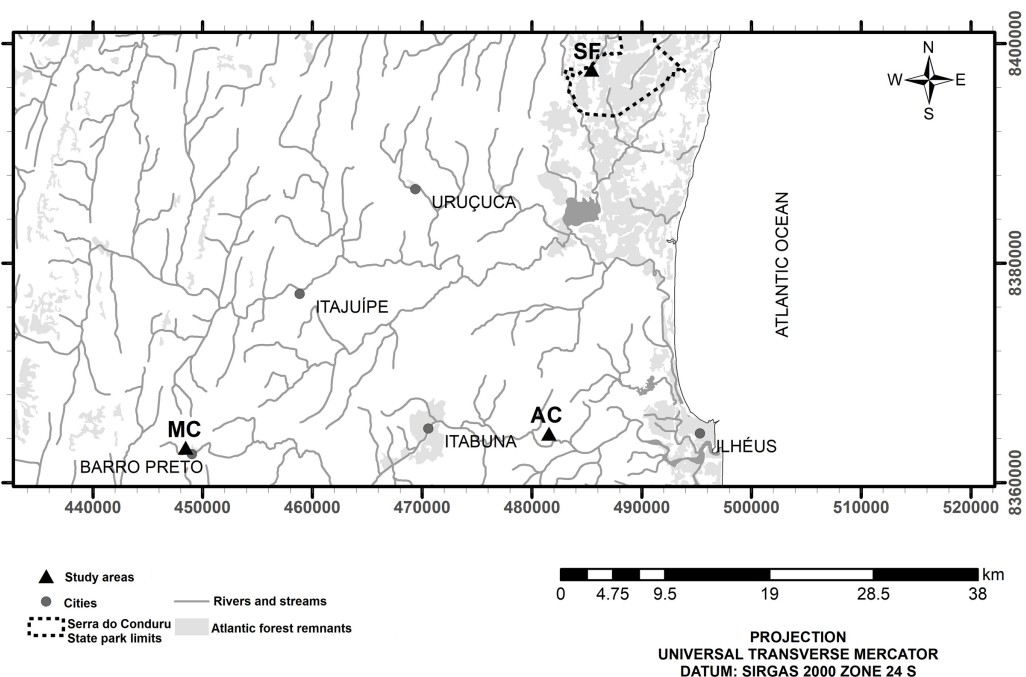

**Figure 1** **Location of study sites in secondary forest (SF), a managed AFS (MC) and an abandoned AFS (AC) in northeastern Brazil.** Map database: Instituto Brasileiro de Geografia e Estatística (Brazilian territory, cities and hydrography, 2017); Ministério do Meio Ambiente (Serra do Conduru State Park limits, 2012), Fundação SOS Mata Atlântica and Instituto Nacional de Pesquisas Espaciais (Atlantic Forest remnants, 2016).                                             

We expected that (i) managed and abandoned cocoa AFS produce more litter than secondary forest where forest structure (*Curvelo et al., 2009*; *Dawoe, Isaac & Quashie, 2010*; *Fontes et al., 2014*) and soil carbon stocks differ (*Gama-Rodrigues et al., 2010*, *Costa et al., 2018*) and nutrients are rapid cycled (*Nair et al., 1999*); (ii) streams running through forests with high litter production tend to receive larger amounts of litter (*França et al., 2009*; *Gonçalves Júnior et al., 2014*), resulting in greater litter standings stocks in the streambeds (*Webster et al., 1994*; *Lisboa et al., 2015*); and (iii) seasonal patterns of litter inputs and standing stocks reflect precipitation patterns because water availability controls litter production (*Tonin et al., 2017*).

# METHODS

## Study area

The study was conducted in the riparian zones of three small watersheds (Fig. 1) representing secondary forest (E 485415, N 8397615), abandoned cocoa AFS (E 481551, N 8364478), and managed cocoa AFS (E 448466, N 8363187). All sites are located in the Atlantic Forest of southern Bahia in northeast Brazil. The climate is wet tropical (hot and humid with no defined dry season, Af according to the Köppen classification) with annual rainfall ranging from 1,100 to 2,200 mm. The study streams are second-order according to the Strahler classification. Daily rainfall data were obtained from the website of the Real-Time Climate Monitoring Program of the Northeast Region (PROCLIMA,

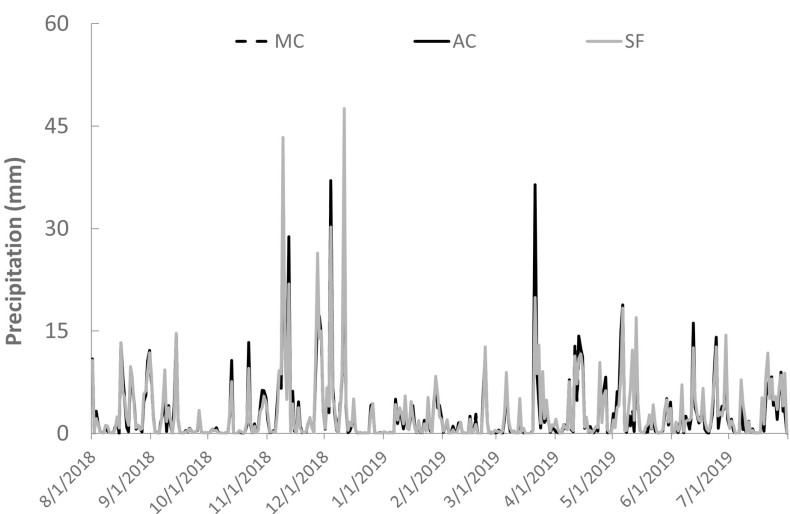

**Figure 2** Daily precipitation at the study sites in secondary forest (SF), a managed AFS (MC) and an abandoned AFS (AC) in northeastern Brazil.

Schwetzingen, Germany; http://proclima.cptec.inpe.br) for the municipalities of Itacaré, Ilhéus, and Barro Preto (Fig. 2).

The secondary forest, which covers 9,275 ha, is located in a conservation area (Serra do Conduru State Park-License 2017-013654/TEC/PESQ-0014) (*Martini et al., 2007*). The vegetation is a mosaic of different developmental stages, including secondary forest and remnants of mature forests with different degrees of selective logging in the past (*Winbourne et al., 2018*). The uniform canopy of the forest exceeds 25 m in height and includes a few emerging individual trees, epiphytes, large lianas, and a dense understory (*Martini et al., 2007*; *Costa et al., 2018*). Tree species density levels in the area were high at all sites, independent of forest successional stage; old growth forest totaled 144 species, old logged forest had 137 species, and recently logged forest 134. Of the species recorded in the Serra do Conduru State Park, 51.4% are endemic to the Atlantic Forest and 26% occur only in the south of Bahia (*Martini et al., 2007*). The abandoned AFS covers 73.4 ha and is located in an AFS (Santa Cruz, CA, USA) where crop management was abandoned 20 years before the present study. Old cocoa trees and other, irregularly distributed species such as jackfruit, erythrina, embaúba, and jequitibá trees (*Argôlo, 2009*) resulted in a medium level of shading (70%). The managed AFS is located in another AFS (Nova Harmonia) with a total area of 89.8 ha. It comprises areas under cocoa production, a forest patch in the central portion, and two areas undergoing regeneration (*Santos et al., 2016*). Management consists of pruning cocoa trees every 6 months, with the biomass left in place (*Costa et al., 2018*), complemented by vegetation cutting, and some liming for soil amelioration. The cocoa plants were spaced at 3 m × 3 m and intercropped with introduced shade trees (erythrina), according to the proposed management for the area.

**Litter inputs and benthic standing stocks**

Litter inputs and benthic standing stocks were determined from August 2018 to July 2019. Details of the methodology are described in *Gonçalves Júnior et al. (2014)* and *Bambi et al.*

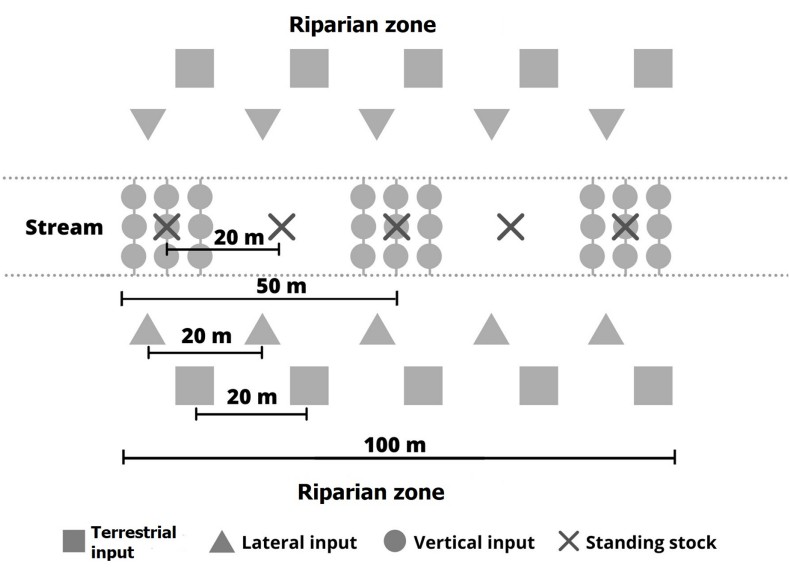

**Figure 3 Sampling design to determine litter inputs and standing stocks in riparian zone and streams.**

*(2016)*. Terrestrial litter fall (TI), vertical (VI) and lateral (LI) litter inputs to streams, and litter deposited on the streambeds (benthic standing stock–BS) were assessed along 100 m stream stretches at each location (Fig. 3).

TI deposited on the riparian soil represents the amount of litter that can potentially be transported to the stream. It was collected with 10 nets (1 mm mesh, 2.5 m² total area), five on both sides of the streams, installed 1 m above the ground at 20 m distance from one another in the riparian zone. VI represents litter that falls directly into the streams from the riparian canopy. It was collected with 27 buckets (30 cm diameter, 1.9 m² total area) fixed to trees, perpendicular to the stream channel at a height of approximately 2 m. The buckets were arranged in three groups of nine, spaced approximately 30 m apart with a distance of 1 m between the individual buckets. Small holes in the bottom of the buckets allowed any collected water to drain. LI represents the indirect input of litter by lateral movement from the forest floor to the stream due to gravity, runoff, wind, or animal action. LI was collected with 10 nets (1 mm mesh, 0.5 m length, 1.5 m² total area) arranged at ground level along the stream margins, five on both sides of the streams. Total litter input to the streams was calculated as the sum of lateral and vertical inputs. Finally, benthic standing stocks represent the litter accumulated on the streambed. It was estimated by taking Surber samples (0.25 mm mesh, 0.45 m² total area), five in each stream at 20 m distance from one another (Fig. 3).

The litter trapped in the nets and buckets was collected at monthly intervals and sorted into four fractions upon return to the laboratory: leaves, branches (*i.e.*, woody pieces less than 25 cm in length), reproductive organs such as flowers and fruits, and miscellaneous material (*i.e.*, unidentified plant matter and animal remains). The sorted litter was dried in an oven at 60 °C for 72 h and weighed. TI, VI and LI were expressed in g dry mass m$^{-2}$ d$^{-1}$, LI per m² was calculated by dividing the collected litter mass by the trap width and

**Table 1 Absolute monthly inputs and standing stocks (g m$^{-2}$) as well as relative contributions (%) of various litter fractions to litter inputs and standings stocks in streams and riparian zones.**

| Litter fraction | Forest type | Terrestrial | | | Vertical input | | | Lateral input | | | Standing stock | | |
|---|---|---|---|---|---|---|---|---|---|---|---|---|---|
| | | Mean (g m$^{-2}$) | Range (g m$^{-2}$) | Contribution (%) | Mean (g m$^{-2}$) | Range (g m$^{-2}$) | Contribution (%) | Mean (g m$^{-2}$) | Range (g m$^{-2}$) | Contribution (%) | Mean (g m$^{-2}$) | Range (g m$^{-2}$) | Contribution (%) |
| **Leaves** | SF | 48 | 0–76 | 65 | 27 | 0–122 | 67 | 18 | 0–164 | 74 | 153 | 6–483 | 36 |
| | MC | 50 | 0–193 | 56 | 30 | 0–194 | 41 | 28 | 0–441 | 65 | 91 | 0–422 | 60 |
| | AC | 64 | 4–322 | 67 | 56 | 0–287 | 69 | 12 | 0–5,160 | 80 | 182 | 30–424 | 61 |
| **Branches** | SF | 6 | 0–92 | 8 | 3 | 0–201 | 8 | 3 | 0–83 | 13 | 81 | 0–431 | 19 |
| | MC | 2 | 0–48 | 3 | 3 | 0–137 | 4 | 2 | 0–103 | 5 | 10 | 0–95 | 7 |
| | AC | 12 | 0–75 | 12 | 6 | 0–86 | 7 | 1 | 0–99 | 7 | 48 | 0–386 | 16 |
| **Reproductive parts** | SF | 3 | 0–32 | 4 | 1 | 0–33 | 2 | 0 | 0–15 | 0 | 7 | 0–76 | 2 |
| | MC | 25 | 0–398 | 28 | 33 | 0–493 | 46 | 7 | 0–389 | 16 | 23 | 0–289 | 15 |
| | AC | 6 | 0–73 | 6 | 7 | 0–306 | 9 | 1 | 0–62 | 7 | 2 | 0–57 | 1 |
| **Miscellaneous litter** | SF | 17 | 0–137 | 23 | 9 | 0–108 | 23 | 3 | 0–63 | 13 | 185 | 0–1,432 | 43 |
| | MC | 12 | 0–217 | 13 | 7 | 0–137 | 9 | 6 | 0–293 | 14 | 28 | 0–395 | 18 |
| | AC | 14 | 0–46 | 15 | 13 | 0–56 | 15 | 1 | 0–47 | 7 | 67 | 0–382 | 22 |
| **Total** | SF | 74 | 8–221 | – | 40 | 0–230 | – | 24 | 0–215 | – | 425 | 28–2,050 | – |
| | MC | 89 | 0–511 | – | 69 | 0–638 | – | 43 | 0–816 | – | 152 | 0–559 | – |
| | AC | 96 | 4–339 | – | 82 | 0–521 | – | 15 | 0–303 | – | 299 | 78–726 | – |

**Note:**
Secondary Forest (SF), Managed AFS (MC) and Abandoned AFS (AC).

multiplying the result by two (to account for inputs from both stream banks) and by the mean channel width (*Pozo et al., 2009*). The annual litter inputs to the streams and riparian zones corresponds to the sum of the mean monthly litter inputs during the study year (Table 1).

## Statistical analysis

Differences among the forests in litter inputs and benthic standing stocks were assessed by generalized linear mixed-effects models (*glmer* function in the *lme4* package of R) with forest type (=site), time and the interaction of forest type and time as predictive variables (*Bates et al., 2015*). We considered trap and time as random factors to account for the pseudoreplicated design of the study, since the three forest types were represented only by a single site each. Separate models were run for each of the plant organic matter fractions (leaves, branches, reproductive organs, miscellaneous material). *P*-values were obtained by likelihood ratio tests (chi-square distribution) of the full model against a partial model without the explanatory variable. All models were tested for error distribution by using the *hnp* package and function in R, and corrected for over-or underdispersion. Differences in litter inputs and standing stocks among forest types (sites) were also assessed by using bootstrapped 95% confidence intervals, which were computed by the bias-corrected and accelerated (BCa) method using the *boot* package and function in R, based on 1,000 bootstrap replicates (*Davison & Hinkley, 1997*; *Canty & Ripley, 2016*). Differences were considered statistically significant when the bootstrapped confidence intervals did not overlap.

Given their flexibility, generalized additive mixed models (GAMM) were used as an additional approach to explore the seasonal patterns of litter inputs (*i.e.*, vertical, lateral and terrestrial inputs) and standing stocks (*Tonin et al., 2017*, *2019*). The input of leaves, branches, reproductive matter, or miscellaneous material over the 12-months study period was used as a normally distributed (identity-link function) predictor, nested within sites, as a random component of the GAMM models. Trap and time were used as random factors in the GAMM models. The degree of smoothing in an additive model is expressed as effective degrees of freedom (edf). Higher edf values indicate a lower degree of linearity (*i.e.*, here variation over time), with a value of 1 indicating a perfectly linear effect. The additive mixed models were fitted by using the *by* command in the *mgcv* package in *R*. Validation was used to estimate the optimal degree of smoothing (*Wood, 2017*). The residual spread within models among sampling dates was measured by using the *varIdent* function in *R*.

## RESULTS

Leaf material was the single largest litter fraction, with percentages >60% for TI and VI and >70% for LI in the secondary forest and abandoned AFS (Fig. 4, Table 1). In the managed AFS, the percentages of TI, VI, and LI were 56%, 41%, and 65%, respectively. Leaves also represented large portions of the instream standing stock of litter, 61% in the abandoned AFS, 60% in the managed AFS, and 36% in the secondary forest. Miscellaneous types of organic matter was the second most abundant litter fraction of TI and VI observed in secondary forest and abandoned AFS (23% and 15% for TI and VI). Miscellaneous litter accounted for 43% of the standing stock in the secondary forest (Table 1). Branches were also a large portion of the litter standing stock in secondary forest (13%) and abandoned AFS (7%) (Table 1). Because of their sporadic and transient occurrence, reproductive parts were a low proportion of the total litter, except in managed AFS (Table 1), where they accounted for a higher proportion in all types of litter inputs and standing stocks (Figs. 4C, 4H, 4M and 4R). Branches and miscellaneous litter were generally similar in all forests, except for higher benthic standing stocks of miscellaneous litter in the secondary forest (Fig. 4).

The greatest seasonal variation in litter inputs and standing stocks was found for leaves, with the highest contributions during the transition between dry and rainy months and two seasonal peaks in some cases (Figs. 5A and 5C). Seasonal variation was less pronounced and generally not significant for branches, reproductive plant parts, and miscellaneous litter (Figs. S1 to S3). Vertical litter inputs showed seasonal patterns for leaves in the secondary forest (effective degrees of freedom—edf = 5.5; Fig. 5D) and abandoned AFS (edf = 5.1; Fig. 5E), with higher contributions during the rainy months, from November to February, in both SF and AC. In managed AFS, the contribution of leaves decreased linearly over time, as reflected by an edf value close to 1 (Fig. 5F). Terrestrial leaf inputs showed a sinusoidal pattern is the secondary forest (Fig. 5A) and managed AFS (Fig. 5C), which was reflected by high edf values of 7.0 and 6.7, respectively. A peak in the rainiest months was observed in all three forests but the second peak was missing in the abandoned AFS (Fig. 5B). The largest lateral inputs of leaves in managed

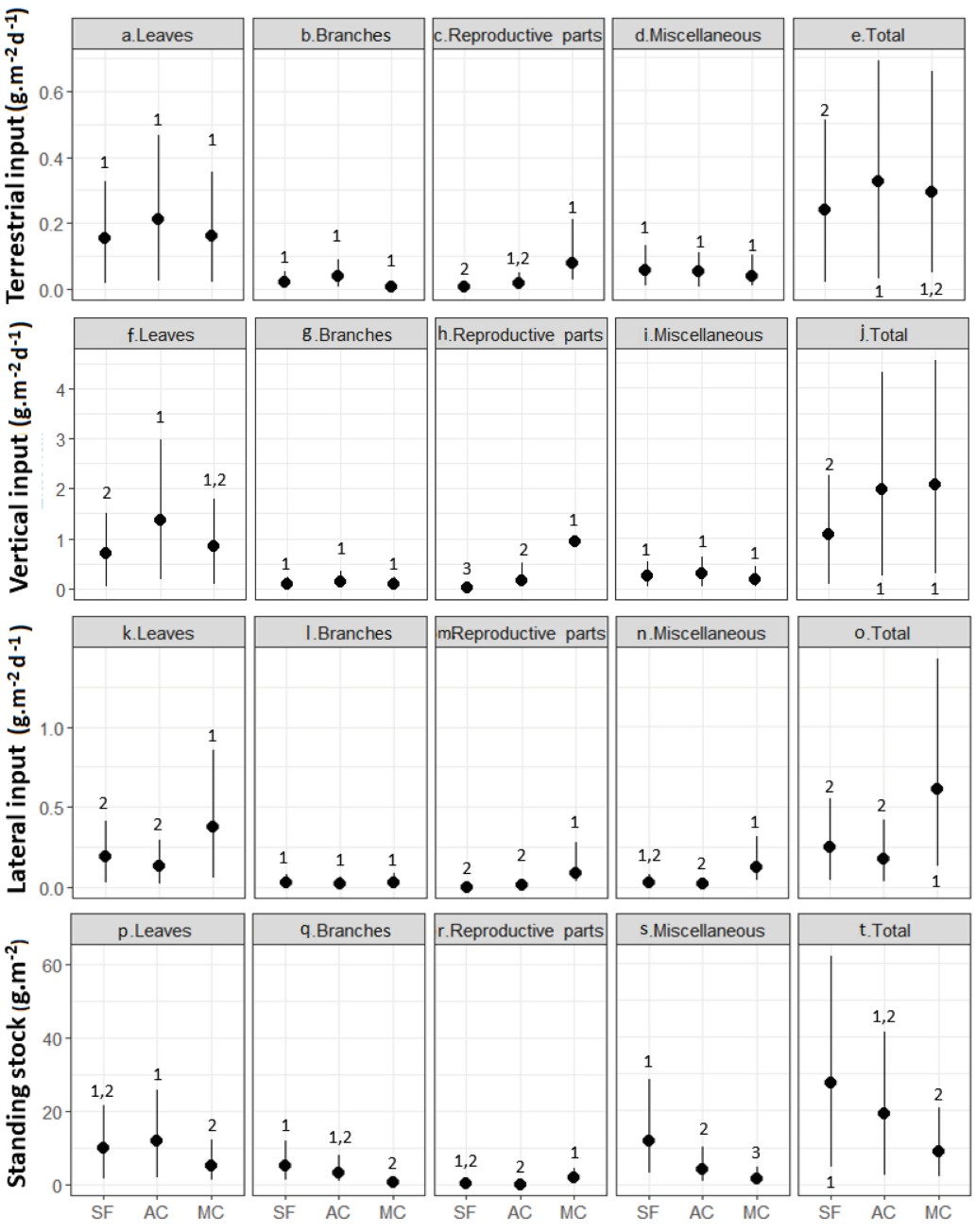

**Figure 4 (A–T) Inputs and standing stocks of various litter fractions in streams and riparian zones of secondary forest (SF), a managed AFS (MC) and an abandoned AFS (AC).** Black circles and vertical lines represent means and bootstrapped 95% confidence intervals. Different numbers indicate significant differences of means as judged based on non-overlapping confidence intervals.

AFS were observed from January to March (edf = 1.9; Fig. 5F), the period of least rainfall (Fig. 2). Standing stocks of leaf litter showed different trends than leaf litter inputs. The sine curve shifted to the right in SF, indicating that leaf input occurred just after the rainiest periods (peak in April; edf = 3.5; Fig. 5J) and abandoned AFS (minimum in

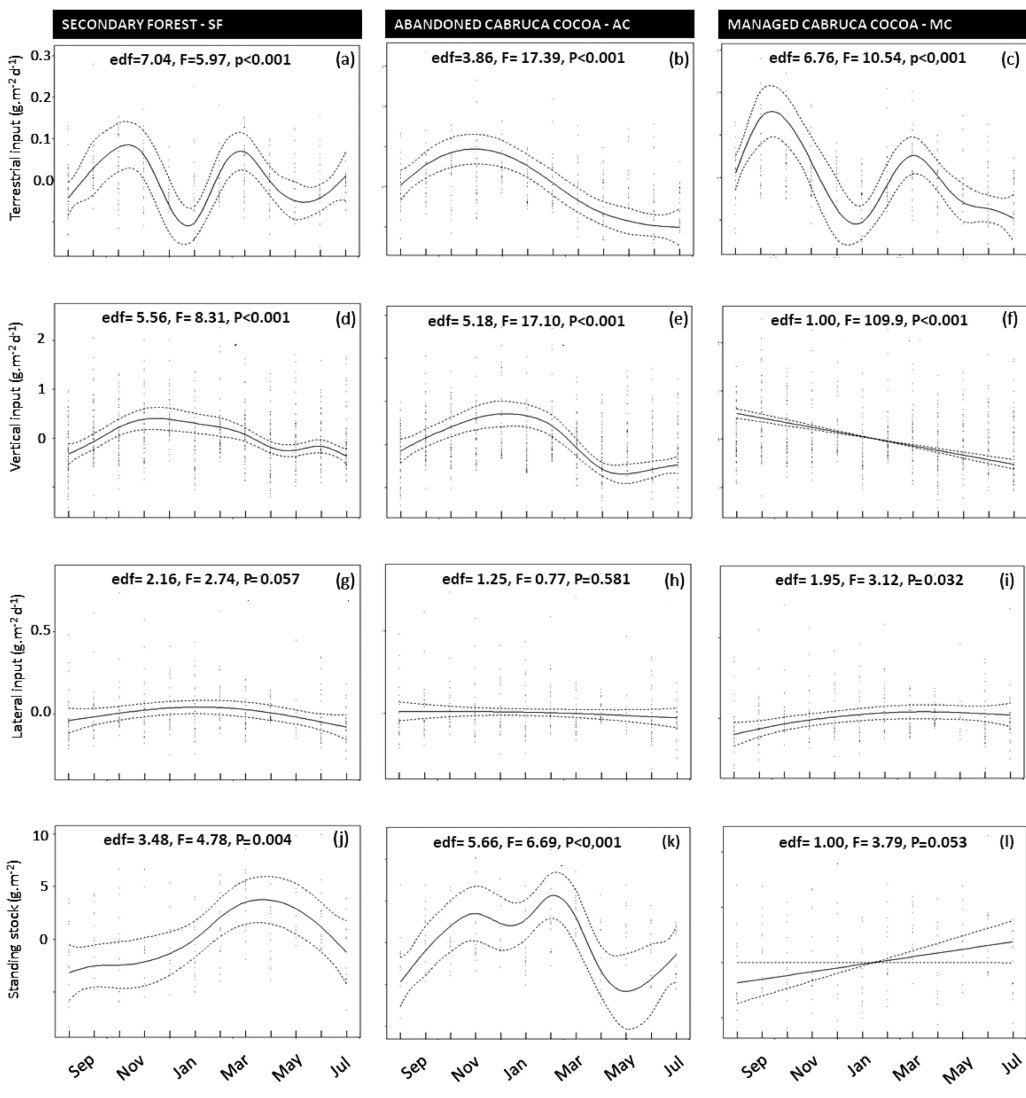

**Figure 5 (A–L) Temporal changes of litter inputs (g dry mass m⁻² d⁻¹) and standing stocks (g dry mass m⁻²) in secondary forest (SF), an abandoned AFS (AC) and a managed AFS (MC).** Also shown are F and P values as well as the effective degrees of freedom (edf) of GAMM analyses. Continuous lines are the GAMM smoothers and dotted lines indicate 95% confidence limits.

November and maximum in February; edf = 5.6; Fig. 5K). No seasonal patterns were observed for leaf litter of managed AFS (edf = 1; Figs. 5F, 5I and 5L).

Total annual litter fall in the riparian zone of the abandoned AFS, managed AFS, and secondary forest was 1,889, 1,016, and 1,176 g m⁻² yr⁻¹, respectively. In the abandoned and managed AFS, 56% of the terrestrial litter input (TI) was deposited on the forest floor and 44% directly entered the streams through vertical litter input (VI). In the secondary forest, the percentages of TI and VI were 63% and 37%, respectively. In the managed AFS, 61% of the total litter fall in the riparian zone entered the stream by lateral movement and 39% remained in the riparian zone. In the abandoned AFS, only 15% entered the stream and 85% remained in the riparian zone. The corresponding percentages in the secondary forest

were 30% and 70%. The average annual total litter standing stock in the managed AFS was more than two times lower than in the abandoned AFS and more than three times lower than in the secondary forest (Fig. 6).

## DISCUSSION

Riparian zones with natural vegetation are important for the structural and functional integrity of streams (*Gonçalves Júnior et al., 2014*, *Rezende et al., 2017b*). Our results on the contribution of leaf litter in forests subject to different management practices are important information to assess the role of cocoa agroforestry in efforts to restore impacted forests in northeastern Brazil. Previous studies in the area have reported that the cocoa agroforestry system alters the biogeochemistry of C and N in streams and soils (*Costa et al., 2017, 2018*; *Souza et al., 2017*). However, these studies could not determine which forest attributes determined the observed changes in C and N cycling. Although differences between forests could be associated with different management regimes, it is necessary also to consider the potential influence of other factors that could not be evaluated in those study.

High litter production in the abandoned AFS may be due to the riparian vegetation structure (*Gonçalves Júnior et al., 2014*; *Rezende et al., 2017a*) and successional stage during forest recovery (*Sambuichi & Haridasan, 2007*; *Rolim et al., 2017*). Factors such as abundant deposits of crop biomass (*Beer et al., 1998*), which are related to high carbon stocks in the soil (*Gama-Rodrigues et al., 2010*; *Costa et al., 2018*), and rapid nutrient cycling in these systems (*Nair et al., 1999*) are likely to play a role. The pattern of litter inputs in the abandoned AFS was more similar to that in the secondary forest than the managed AFS. Streams in abandoned AFS are usually lined by riparian vegetation (*Ferreira et al., 2019*) similar to that of streams in secondary forest, whereas in managed AFS, native shade trees are replaced by species with high commercial value (*Cassano et al., 2009*, *Piasentin, Saito & Sambuichi, 2014*). Additional factors linked to higher litter production in abandoned AFS (*Gama-Rodrigues et al., 2010*) include greater stand age of the vegetation. Moreover, leaf surface area may be greater in the shaded stands where low solar radiation limits rates of photosynthesis (*Beer et al., 1998*). These factors tend to increase litter production, to which leaves contribute the largest fraction (*Gonçalves Júnior et al., 2014*; *Bambi et al., 2016*; *Tonin et al., 2017*).

Higher litter production in the abandoned AFS is reflected in the observed spatial and seasonal patterns of litter inputs and standing stocks, which were also closer to those in the secondary forest than in the managed AFS. This indicates that abandoned AFS with little human intervention may provide favorable conditions for forest regeneration (*Rolim et al., 2017*) and could thus be valuable strategic sites for the conservation of riparian forest remnants in these agroforestry systems of Brazil (*Faria et al., 2007*; *Cassano et al., 2009*; *Schroth et al., 2011*; *Sambuichi et al., 2012*). The presence of pioneer species in abandoned AFS could be a critical factor promoting this regeneration by facilitating ecological succession (*Rolim & Chiarello, 2004*, *Sambuichi & Haridasan, 2007*; *Sambuichi et al., 2012*).

SF

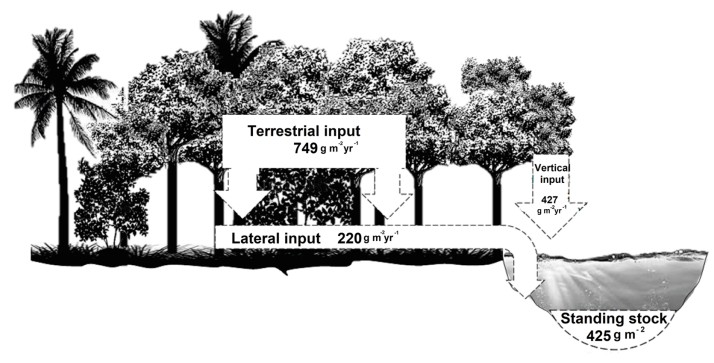

AC

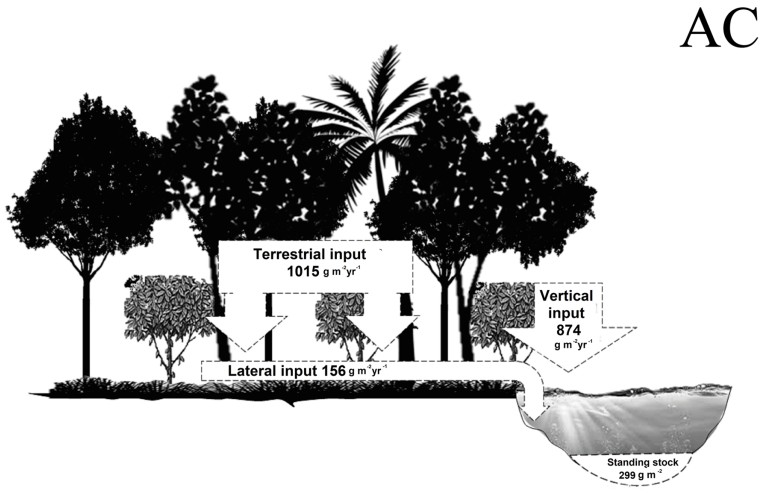

MC

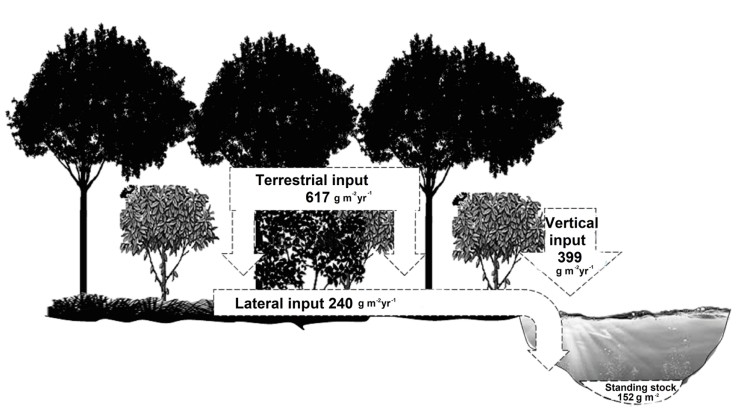

**Figure 6 Summary of annual litter inputs and average standing stocks (g dry mass m⁻²) in streams and riparian zones.** SF, secondary forest, AC, abandoned AFS and MC, managed AFS.

Crop management, which involves hoeing and soil cleaning (*Sambuichi et al., 2012*; *Mello & Gross, 2013*), may be a critical factor accounting for the observed tendency of greater lateral litter inputs to streams in managed AFS. Possible mechanisms include facilitation of litter leaching and movement of litter along streams by runoff (*Afonso, Henry & Rodella, 2000*; *Wantzen et al., 2008*) as well as effects on the structure and composition of the tree vegetation (*Deheuvels et al., 2014*). Furthermore, some management practices introduce exotic species necessary for cultivation (*Sambuichi, 2002*; *Piasentin, Saito & Sambuichi, 2014*; *Rolim et al., 2017*) and involve thinning of the vegetation to obtain the desired shade levels for crop production (*Johns, 1999*).

The management practices in cocoa cultivation and phenology of the shade species may also have determined the greater contribution of reproductive plant parts in the managed AFS. This observation is related to the high proportion of exotic species (*e.g., Artocarpus heterophyllus*, *Spondias mombin* and, particularly in this study, *Clitoria fairchildiana*), which show a different phenology than native riparian forests (*Sambuichi & Haridasan, 2007*; *Sambuichi et al., 2012*). Furthermore, variation in the size, shape, texture and anatomy of reproductive plant parts of different tree species in different forest types could account for the higher contribution of this litter fraction in managed AFS. This applies particularly to fruits of the legume *Clitoria fairchildiana*, the species contributing most to the reproductive plant parts in litter vertical inputs and standing stocks in this forest. The large dry and dehiscent fruits of the tree are 25 to 30 cm long and 2.6 to 2.9 cm wide (*Silva & Moro, 2008*). A potentially higher nutritional quality of such reproductive plant parts compared to leaves may favor rapid decomposition of the litter supplied to tropical streams. However, the fruits of *Clitoria fairchildiana* and other species in the managed AFS we investigated may not provide a high-quality nutritional resource (*Rezende et al., 2017c*), and we do not currently have data to evaluate the consequences for litter decomposition.

Litter standing stocks in ecosystems are the net result of litter inputs and losses by movement and decomposition (*Elosegi & Pozo, 2005*; *Tank et al., 2010*). The high standing stocks we observed in streams of secondary forests could imply high inputs (*França et al., 2009*; *Lisboa et al., 2015*; *Bambi et al., 2016*) combined with slow decomposition and downstream transport (*Gonçalves Júnior et al., 2014*; *Rezende et al., 2017a*), and the high proportion of miscellaneous matter in the secondary forest stream could be due to the particularly slow decomposition of this organic matter fraction. Effective litter retention favors long residence times in streams (*Bilby & Likens, 1980*) and the generation of small organic particles released during decomposition (*Gessner, Chauvet & Dobson, 1999*, *Boyero et al., 2011*) instead of coarse litter being transported downstream. This contrasts with the situation in the managed AFS where cocoa leaves are the predominant litter type, the high lignin and cellulose concentrations of which slow litter decomposition in the soil of cocoa AFS (*Dawoe, Isaac & Quashie, 2010*) and likely also in streams (*Tank et al., 2010*; *da Silva et al., 2017*). In the managed AFS, in contrast, it may have been the high proportion of recalcitrant reproductive plant parts (see above) that favored high benthic standing stocks. This type of litter was rare in the secondary forest and abandoned AFS.

Our finding that litter inputs and standing stocks in the abandoned AFS were more similar to those in the secondary forest than in the managed cocoa AFS suggests that

abandoning management measures, such as litter removal and soil cleaning, could create favorable conditions for reestablishing natural litter dynamics in riparian zones and streams of former AFSs. The capacity of tree species richness to regenerate is high in those forests (*Sambuichi & Haridasan, 2007*), if surrounding vegetation remains intact to provide a seed source for forest recovery (*Rolim et al., 2017*). It appears that the absence of management in the riparian zone of abandoned AFS sufficiently reduces pressure on species during regeneration (*Rolim et al., 2017*) for the phenology of species to become the main factor determining litter dynamics. Intensive management, in contrast, overrides the importance of phenology by altering the structure of riparian plant communities (*Delong & Brusven, 1994*; *Ferreira et al., 2019*).

## CONCLUSION

Although our study was restricted to one location for each of the three investigated forest types (SF, AC and MC), our findings are a starting point to evaluate differences in litter inputs and standing stocks between those forests. The observed similarity with litter inputs and standing stocks in the secondary forest suggests potential of the abandoned AFS to provide favorable conditions for restoring natural litter dynamics in streams and riparian zones. However, future investigations are needed to elucidate the nutritional quality of litter and its variation depending on the composition and structure of the riparian vegetation.

## ACKNOWLEDGEMENTS

We are grateful to Mr. Hermann Rehem for granting us permission to collect data on his farm. We also appreciate the support and partnership of the Aquariparia research group at the Universidade de Brasilia–UnB. Finally, we thank Ms. Cipriana Leme for language editing.

### Funding
This work was supported by UESC (PROPP register 0220.1100.1771), CNPq (403945/2021-6), and FAPESB scholarship (TO 715/2017). The funders had no role in study design, data collection and analysis, decision to publish, or preparation of the manuscript.

### Grant Disclosures
The following grant information was disclosed by the authors:
UESC: 0220.1100.1771.
CNPq: 403945/2021-6.
FAPESB Scholarship: TO 715/2017.

### Competing Interests
The authors declare that they have no competing interests.

## Author Contributions

- Haialla Carolina Rialli Santos Brandão conceived and designed the experiments, performed the experiments, analyzed the data, prepared figures and/or tables, authored or reviewed drafts of the article, and approved the final draft.
- Camila Andrade Coqueiro Moraes performed the experiments, analyzed the data, prepared figures and/or tables, and approved the final draft.
- Ana Paula Silva performed the experiments, prepared figures and/or tables, and approved the final draft.
- José Francisco Gonçalves Júnior conceived and designed the experiments, analyzed the data, prepared figures and/or tables, authored or reviewed drafts of the article, and approved the final draft.
- Renan de Souza Rezende analyzed the data, prepared figures and/or tables, authored or reviewed drafts of the article, and approved the final draft.
- Daniela Mariano Lopes da Silva conceived and designed the experiments, analyzed the data, prepared figures and/or tables, authored or reviewed drafts of the article, and approved the final draft.

## Field Study Permissions

The following information was supplied relating to field study approvals (*i.e.*, approving body and any reference numbers):

Field experiment in Serra do Conduru State Park was approved by INEMA (Process 2017-013654/TEC/PESQ-0014).

## Data Availability

The raw data are available in the Supplemental Files.

## Supplemental Information

Supplemental information for this article can be found online at http://dx.doi.org/10.7717/peerj.13787#supplemental-information.

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
