# Peer review of "Litter inputs and standing stocks in riparian zones and streams under secondary forest and managed and abandoned cocoa agroforestry systems"

_PeerJ, doi:10.7717/peerj.13787_

## Round 0.1 · original submission · Major Revisions

Dear Dr. Rialli Santos Brandão

We have now received two cogent reviews of your manuscript. Both see merit in the study but also point out that the presentation is well below standard, to the point even that a detailed scientific evaluation is difficult. Reviewer #2, in particular, makes excellent recommendations on how to improve the manuscript. Please make a particular effort to adhere to them. Reviewer #1 has similar concerns and has also written numerous comments directly in the manuscript. Please note, however, that it will not be sufficient just to adopt those changes. Instead, a fundamental revision will be necessary. I will only be able to send the manuscript to review again, if the structure and presentation of the manuscript have been massively improved. This concerns all aspects of the presentation and writing, from the organization of ideas to word choice, grammatical and typographical correctness, format of the references, quality of the figures, and overall coherence.

The lack of data on primary forest, highlighted by Reviewer #1, to provide a baseline is a problem, as is the lack of true replication, which Reviewer #2 correctly pointed out. However, I understand that it will not be possible to address these limitations at this stage. Or do you have additional data?

You do not need to worry about the criticism expressed by Reviewer #1 relating to the novelty of the study. Novelty is not a criterion that PeerJ uses in its evaluation procedures.

In your response letter please indicate point by point how you have dealt with the criticism. Please also confirm that the manuscript has been properly edited, preferably by a native speaker with experience in editing ecological/scientific papers, as pointed out by Reviewer #2.

I look forward to receiving your thoroughly revised manuscript in due course.

Best regards

Mark Gessner


PS: In case the PDF file annotated by Reviewer #1 is not appended, please contact the editorial office.

·

Basic reporting

The Atlantic Rainforest of Brazil is a fragile and vanishing ecosystem. Advancement of management practices that support it are clearly needed. This study examined the contribution of cabruca cocoa agroforestry practices to conservation and economic development in the region. Unfortunately I felt that the paper gave me more questions than answers.

The paper was generally poorly written, with poor expression, many grammatical and typographical errors, and the meaning was often unclear. More field background information and context is needed to give a more thorough understanding of the cabruca cocoa management practices and what happens when the plantations are abandoned, or in comparison to secondary forest. Furthermore, to fully understand the conservation implications, I would like to know how they compare with primary forest. The first sentence mentions the potential for biodiversity conservation, yet biodiversity was not measured in any form in the study. I think an understanding of the change in plant diversity between the three systems is needed to support the conclusions of the study.

The hypotheses were not clearly introduced and explained. What led you to come up with these hypotheses?

Experimental design

The basic design of litter measurement is fine, but I don't think that litter tells enough of a story - more information is needed. e.g. plant diversity and/or litter decomposition would provide a greater understanding of the three ecosystems.

A comparison with primary Atlantic rainforest would also be helpful - even if this was just regarding plant diversity based on the literature.

Validity of the findings

I didn't find the study to be particularly impactful. The results were extrapolated beyond their validity given that the only factors examined were litter inputs over time.

The results were over-described. The figures tell the story.

Although literature sources were given to support the statement ‘Thus, the importance of cabruca cocoa AFS contribution to the local landscape of southern Bahia, Brazil, is emphasized, which play a significant role in the conservation of regional biodiversity’, it is not actually supported by this study, given that it was only about the quantity of litter inputs.

Additional comments

I have attached a pdf of your paper with comments, suggestions and errors highlighted. Most of the comments are requests for clarity. I hope they will be helpful. I think the paper would benefit from careful editing.

Reviewer 2 ·

Basic reporting

The manuscript of Brandão et al. entitled “Litter dynamics in riparian zones of the Atlantic Rainforest under cocoa agroforestry” presents the results of a descriptive study developed in northeast Brazil to evaluate the dynamics of organic matter in 3 streams with different land uses in the riparian zones: managed cocoa agroforestry systems, abandoned cocoa agroforestry systems and secondary forest. Studies focused on the dynamics of organic matter in agroforestry catchments are rare and the authors obtained interesting results. However, the current version of the manuscript has several weaknesses that will require substantial work before publication.

The text is well referenced but the contextualization and relevance of the study to an international audience must be better emphasized in the text. In addition, the English language is substandard to a research article. Several sentences are not completely clear and all acronyms used should be carefully revised because some of them came from words in Portuguese that do not make sense in English. I could also find many typos throughout the text. Because none of the authors is a native English speaker, I think the whole text should be carefully revised by a language professional editor.

Because authors used different styles of citations in the text, I think the manuscript is not formatted according to the PeerJ style. I also think the number of figures is excessive for a research article. In addition, I suggest the inclusion of Table S1 in the main document of the manuscript.

Experimental design

In spite of this experimental design was already used in other studies, I am concerned because different traps were used to determine the different types of litter inputs and benthic stock. Even with the conversions of the obtained data to m2, the sampling effort differs among traps and this may result in some bias in data interpretation. The lack of replicates, i.e. only one stream was studied in each land use condition, is another concern that should be better justified in the text. Because this is a 1-year study, authors must interpret the annual variation with caution (lack of temporal replicates).

Validity of the findings

The obtained data is robust and the statistical approach used is in accordance with the current literature. However, I suggest authors rethink their strategy for presenting the results, to avoid the presentation of raw data. As it stands, the Results chapter is also long and makes difficult the reader’s comprehension.

Additional comments

The abstract is relatively well written but the hypotheses of the study are missing. The main results and the conclusions of the manuscript should be rewritten to make the text clearer.

The Introduction chapter is quite long and some relevant topics used in the premises of the study are missing (e.g., the effects of temperature and precipitation on litter production). In addition, the objective, premises, and hypothesis of the study are not well stated. I think authors should reorganize the whole Introduction to make it more clear, focused, and shorten.

The Materials and Methods should also be reorganized because (i) the description of the catchments studied is long and excessively detailed and (ii) the presentation of the experimental design was omitted. Even this design has been used in other studies, the authors should give more details about it in this chapter.

The presentation of the Results is quite chaotic and should be completely reorganized. Authors should present the results in a logical sequence and values must be presented in tables to facilitate the reader’s comprehension and comparisons among the studied catchments. Table S1 is fundamental in this chapter and should be included in the main document. On the other hand, the number of figures is excessive, and authors must decide which part of the results is most important to support their conclusions. All additional results should be moved to the supplementary material.

More ecological thinking is required in the Discussion and the extension of this chapter is also long. Authors should avoid repeating the obtained results. The text of the Discussion should start focusing on the interpretations of the results and became wider to discuss the contribution of this study to the existing understanding of the different types of agroforest systems and their consequences to the dynamics of organic matter and ecosystem biodiversity and processes. A clear conclusion of the study is also missing.

---

## Round 0.2 · Major Revisions

Dear Dr. Brandão

Please accept my apologies for the long delay in responding to the resubmission of your revised paper. As you will see in the attached file, I am afraid that despite this delay the manuscript is not yet ready for publication. Although I agree with Reviewer 2 that the revison has improved the manuscript, I do not share the view that the paper is now satisfactory.

Nevertheless, I have started editing the manuscript to streamline and strengthen the paper. You will see that I used a heavy hand. Please give these changes and queries careful attention, while paying attention to correcting any errors that I could have introduced here and there, partly because the statements were not always clear. I attach a PDF with my comments and changes but will also ask the editorial office to forward you the Word file where I used tracked changes.

Although the great majority of the changes are of an editorial nature, there are also several fundamental concerns that still need attention:

1) It appears you have summed up the litter standing stocks in the streambed instead of calculating an annual average. That needs to be double-checked and corrected.

2) Accordingly, the units for standing stocks must be g/m2 instead of g/m2/d or g/m2/yr and the numerical values need to be corrected in the text, figures and tables, including in Fig. 6.

3) Clarification is needed why the measurement of litterfall with the nets and buckets leads to such different results. If this is not a methodological issue, which I doubt can cause a difference as big as the one observed, the only explanation I have is that the canopies above the streams are not closed. Is that the case? Please provide detailed information in the Study Area section about the situation. What puzzles me is that the streams are second order, which should lead to fully closed canopies. Or is the schematic in Fig. 3 drawn to scale, which would suggest a stream width of 20 m, although that is hard to imagine for a second-order stream. In any case, this issue needs careful explanation.

4) Please also clarify why you chose not to use standard statistics (ANOVA, GLM) to analyse the data, which would have enabled a much more powerful analysis. GAMM is not designed for the type of data you present, but only an unsatisfactory emergency method for this kind of data. If the data severely violate the key assumptions of parametric statistics, you could use PERMANOVA instead.

5) I am concerned that the temporal patterns shown in Fig. 5 are not robust. The reason is that they are different for nearly every panel. That prompts the question of whether the GAMMs adequately capture the situation. At the very least you must show the data points in the graphs rather than just the modeled trends.

6) You also need to alert readers to the fact that your design with just one forest/stream of its kind is a case of pseudoreplication, as has previously been pointed out by the reviewers. You cannot derive statistically meaningful inferences on the difference between forest types from this study. All you can do is compare the three different forests. Any of the observed differences could be due to the different management regimes, yes, but equally well also to any other known or unknown factor or factors. Although you cannot change this point, as I had written earlier, conclusions about the forests types are speculative and must be toned down and qualified accordingly. You simply ignored this criticism in the revised ms.

7) Similarly, you need to address the problem of the lack of data on primary forest, as had also been pointed out before but escaped your attention. Here, too, caution is needed with your conclusion about the value of abandoned AFS.

Except for a few haphazard comments and some deletions here and there, I have not yet edited the Discussion. A lot more needs to be done to make it convincing. Consequently, please thoroughly revise the Discussion, keeping in mind that it is meant to interpret, not (re)present, your data. The interpretation includes pointing out noteworthy implications of your results, but those implications should derive from your results. Avoid motherhood statements.

Finally, I have various points to improve the quality of the figures; I suggest striving to use a consistent format (proportions, fonts etc.), particularly for Figs 4 and 5, but also 3:

Fig. 1
- The technical quality of the photographs is insufficient
- Translate Portuguese to English; this includes using a decimal point, not a comma, and “and” instead of “e”
- Do you mean reservoir instead of lagoon?
- The detailed map is superfluous as far as I can see. A map of Brasil showing the Atlantic Forest and the study locations along with high-quality photographs would be sufficient. The coordinates are given in the text.

Fig. 2
- Label y-axis: Precipitation (mm)
- Remove every second y-axis label (0, 10, 20... 50) and increase font size to allow printing the figure in a single column. This also requires increasing the font size of the dates along the x-axis.

Fig. 3
- Clarify whether the schematic is drawn to scale
- Adjust the legend

Fig. 4
- Arrange panels in consistent order. Normally you start with the traps in the riparian zone (= Terrestrial input).
- Clarify terminology. Consider using the terms “Riparian inputs,” Vertical inputs” and Stream inputs” throughout.
- Put units in parentheses.
- Replace periods in the units by blank spaces.
- Introduce letters for all panels (a to p), so you can refer to them individually, not just to the rows.
- Leaves, branches etc. do not need to be repeated in each row (also remove l.c. letters here).
- Why are the errors bars asymmetrical?
- Remove “d-1” from the unit for standing stocks and correct numerical values accordingly (see general comment above)

Fig. 5
- The figure is blurred. Is this a PDF problem? Otherwise improve.
- Fonts are too small
- Use consistent order of items
- Use format consistent with that of other figures, particularly Fig. 4
- Use AFS instead of cabruca.
- See above for issues relating to content.

Fig. 6
- Remove all digits after the decimal point
- Add CI
- Use consistent terminology with other parts of ms (see above)
- Use consistent fonts (Litterfall to streams is different) and enlarge enough to set figures in a single column of the journal
- See above for issues relating to content, especially the unit and numerical values for the standings stocks.

Finally, I have written various tracked changes in Table 1, which I will also attach as a PDF or ask to have forwarded to you as a Word file.

Reviewer 2 ·

Basic reporting

The manuscript of Brandão et al. entitled “Litter dynamics in riparian zones of the Atlantic Rainforest under cocoa agroforestry” presents the results of a descriptive study developed in northeast Brazil to evaluate the dynamics of organic matter in 3 streams with different land uses in the riparian zones: managed cocoa agroforestry systems, abandoned cocoa agroforestry systems and secondary forest. Studies focused on the dynamics of organic matter in agroforestry catchments are rare and the authors obtained interesting results. The authors reorganized all chapters of the manuscript and addressed all weaknesses that were highlighted in the previous version.

The text is well referenced and now, in the revised version, the contextualization and relevance of the study are clearly presented. In addition, the English language was revised and improved, and the acronyms were standardized. The number of typos reduced considerably but, because some are still present in the text, another careful revision of the writing is recommended.

Experimental design

The experimental design was already used in several studies and, regarding the experimental design and statistical approach, all concerns raised in the first round of revisions were clearly justified or answered. The advantages of using GAMM models were also highlighted in the text.

Validity of the findings

The obtained data is robust and the statistical approach used is following the current literature. The Results were reorganized to avoid the presentation of raw data and the extension of this chapter was substantially shortened.

Additional comments

The chapters of the manuscript were reorganized and several paragraphs reworded. The Results and Discussion, the chapters that needed substantial improvement, are much better in the current version. All main comments raised by the reviewers in the previous version of the manuscript were addressed and/or justified. The number of figures was also reduced.

---

## Round 0.3 · Major Revisions

Dear Dr. Lopes da Silva

Thank you for your revised manuscript. Unfortunately, not all of my previous comments have been appropriately addressed. This includes the problem of reporting standing stock data, despite explanations in the response letter to the contrary. Therefore, please revisit my previous comments, in addition to the new ones I have written directly in the file.

Some of the new comments concern the revised Methods subsection in statistics. Please also ensure that the reported statistical results are correct.

You argue that some of the graphs would be cluttered if the data were shown in addition to trend lines. I understand that there can be quite a bit of scatter. But this is precisely what readers must see to evaluate the data and the conclusions derived from them. Therefore, please add those data points to Figs. 5 and S1-S3.

Finally, I have again written a large number of additional queries and suggestions directly in the manuscript, this time particularly in the Discussion, which I had not closely looked at before. Please give attention to all of those points as well. They include further comments and edits on the figures, tables and their legends, which I have copied to the end of the manuscript file.

I look forward to receiving your revised manuscript in due course, hoping that I can recommend accepting it for publication in PeerJ after those revisions.

Best regards

Mark Gessner


PS: I will ask the Editorial Staff to send you a Word file with tracked changes again.

---

## Round 0.4 · Minor Revisions

Dear Dr. Mariano Lopes da Silva

Thank you for your revised manuscript. Before I can recommend acceptance of your paper, I am afraid that I must again return to four points we have discussed earlier. I am referring below to the numbers you used in your rebuttal letter that I received on 11 March 2022:

3) Re the treatment of “trap” in the GLMs (lines 173.181), I am sorry that I do not understand your response: “Trap represents the compartments, which have been individually tested, in this way trap ever as a main factor.” You state in the manuscript that “Trap” is “a predictive variable” and that you “considered trap and time as random factors.” You also state that “forest type (i.e. site), time and the interaction of trap and time as predictive variables.” Did you mean to say that your factors were forest type (=site) and time and that you also included the interaction of forest type and time, with traps being used as replicates?

Similar queries apply to the GAM models.

I suggest also running a GLM and GAMM for the TOTAL inputs, in addition to the individual organic matter fractions.

5) What do you mean by “hydrological pathway”? Can this be deleted?

9) One solution is to make separate plots for the functions and the data points, then superimpose them. I think it is important for readers to see the actual data, especially with GAMMs, which are so flexible.

11) This is the most important point. It’s not a matter of preference or opinion. Including time in the dimension is simply wrong. Please correct the units and the calculations. Standing stocks are a pool, not a flux. Consequently, the dimensions for the standing stocks (g/m2) and the inputs must be different (g/m2/yr). Unfortunately, there are other published papers that made the same mistake. It must not be repeated. Note that the correct calculation will not change your narrative. And you don’t even have to redo the statistics.

Please also see the comments and tracked changes in the attached PDF, including on the figures and tables.

Best regards

Mark Gessner

---

## Round 0.5 · Minor Revisions

Thank you for the latest revisions. I hope this is the last time I am returning your manuscript. I first thought this would be unnecessary, but please check all the reported values again. Since you report data per day, per month and per year at different places and did not always specify the time unit, I had to guess the correct unit in some instances. So please make sure I got it right. In addition, I was confused by seeing negative values for litter inputs in Figure 5. Please revisit this as well. An annotated PDF file is attached.

---

## Round 0.6 · accepted · Accept

Thank you for your final revision, including the clarification of the negative values in the GAMs. I have no further comments, except that various formatting mistakes slipped in that should be corrected during the production process, and that the new values for total litter inputs (which were extremely low before) are now very high, though not impossibly so. But please double-check the latter once again, just in case. Thank you for submitting to PeerJ.